# Use of Natural Zeolite and Glass Powder Mixture as Partial Replacement of Portland Cement: The Effect on Hydration, Properties and Porosity

**DOI:** 10.3390/ma15124219

**Published:** 2022-06-14

**Authors:** Dalius Kriptavičius, Giedrius Girskas, Gintautas Skripkiūnas

**Affiliations:** Institute of Building Materials, Vilnius Gediminas Technical University, Saulėtekio av. 11, LT-10223 Vilnius, Lithuania; dalius.kriptavicius@vilniustech.lt (D.K.); gintautas.skripkiunas@vilniustech.lt (G.S.)

**Keywords:** Portland cement, clinoptilolite, aluminosilicate, pozzolana, soda lime glass, zeolite suspension, strength, porosity, durability

## Abstract

The study investigates effect of the additive consisting of natural zeolite (clinoptilolite) and soda lime glass powder on the hydration, mechanical properties and porosity of Portland cement concrete. The effect of mineral additive on the technological, physical-mechanical properties and porosity of the mortar was investigated by increasing the content of natural zeolite and glass powder added to the mortar up to 20% by weight of cement in increments of 5% and different particles size of natural zeolite. The mixes with the best technological and mechanical properties were identified and further studies were conducted by replacing 10% and 15% of cement with natural zeolite and soda lime glass with an average grain size of 59.3 μm, 29.0 μm or 3.6 μm of zeolite, and 29.6 μm of glass powder. The hydration process and microstructure of hardened cement paste modified with the aforementioned mineral additives was analysed by microcalorimetry, X-ray diffraction tests and thermogravimetric analysis. The optimal composition of cement paste and particle size distribution of natural zeolite were determined to achieve the higher flexural and compressive strength and lower open porosity. The mixture of mineral additives has the highest effect in terms of flexural and compressive strength and open porosity when added at the proportion 75:15:10 (cement:natural zeolite:soda lime glass) and when zeolite with an average particle size of about 3.6 μm is used

## 1. Introduction

Concrete has very advantageous properties, such as compressive strength, durability, ease of shaping, etc., and therefore it has widespread applications in the construction industry. It goes without saying that the increasing consumption of concrete is directly linked to the increasing consumption of cement, the production of which, in turn, has a significant negative impact on the global ecology due to high emissions of CO_2_ per 1 t of cement (0.73–0.99 t) [1]. The sintering of cement clinkers requires a high temperature, which is reached by burning a large amount of fossil fuel and, consequently, leads to large greenhouse gas emissions. On the other hand, it is very important to reduce the open porosity, which has an undeniably significant negative effect on the durability of cement concretes. It is well known that the higher the absorption capacity of cement concrete, the greater the likelihood of substances, which chemically or physically damage the structure of concrete and thus shorten its service life, to enter the pore system. It is also important to assess the impact of a material on greenhouse gas emissions over its full life cycle, i.e., extraction, production, lifetime and disposal or recycling, as it is clear that the longer the lifetime of a material, the lower the negative environmental impact in terms of GWP (Global Warming Potential).

Nowadays, many researchers are exploring the possibility of using wastes of various materials as inert aggregates or as active additives in the hydration of cement. Much research has been done in an attempt to use crushed concrete, glass, ceramics, bricks, glass, plastics, rubber, etc., waste, by replacing part of a coarse or fine aggregate. Some of these materials may partially reduce the mechanical properties of concrete, but this addresses the issue of waste disposal; others, such as plastic–rubber, can reduce shrinkage deformations [2] and reduce risk of cracks, which increases the durability of the material. Other materials include ground glass, rice husk ash, sugarcane bagasse ash and other non-traditional materials with pozzolanic properties that can react with Ca(OH)_2_ to form compounds very close to C-S-H. On the one hand, this allows some waste to be utilised, and on the other hand, the properties of these materials are used to improve the properties of cement-based composites.

The present paper studies the possibilities of replacing part of the cement with a mixture of natural fossil material (natural zeolite) and production waste (soda lime glass). The advantage of this formulation lies in the ability of one material in the mix to counteract the disadvantages of another material. For example, natural zeolite, having an open crystal structure and high specific surface area, can reduce the separation of free water [3] and, in turn, the workability of concrete [4], whereas the soda lime glass, which does not absorb water, increases not only the workability of concrete, but also the separation of free water [5].

Natural zeolite (clinoptilolite) belongs to the group of tectosilicate minerals and consists of an outer framework of silica and alumina tetrahedra. The framework has a microporous structure formed by open channels of 8- and 10-membered tetrahedral rings. These channels are predominantly occupied by Na, K, Ca and H_2_O [6]. Water molecules and metal cations in the channels participate in the substitution of Si^4+^ by Al^3+^ in the framework [7]. The process of pozzolanic activity of zeolites is not simple and is influenced by a number of factors simultaneously: the surface area, particle size, Si/Al ratio, CEC (Cation Exchange Capacity), mineral and chemical composition, etc. [4]. Zeolite is actually a perfectly crystallised material that changes to an amorphous state during dissolution [8]. The aluminosilicate network of the zeolite starts decomposing in contact with a cement–water mixture under the attack of OH^−^ in a high pH environment [9]. The decomposition mechanism is as follows: in the hydrolysis reaction, the hydroxyl group (OH^−^) weakens the Si-O-Si and Al-O-Si bonds in the surface lattice, and causes the detachment of Si-OH and Al-OH and the subsequent formation of Si-O^−^ and Al-O^−^. The framework with hydrolysed bonds becomes unstable. The weak parts detach and enter the solution [10] where they react with Ca^2+^ and form hydrated calcium silicate and calcium aluminate compounds, very similar to those formed during the hydration of cement [9]. It is widely known that natural zeolites have a high pozzolanic activity. Although the pozzolanic activity of zeolites is much lower than the activity of microsilica, it is higher than the activity of fly ash [11] or granulated blast furnace slag [12].

Literature analysis shows conflicting results for mechanical properties of cement concrete, since some authors indicate that at 28 days the mixes with 15% cement replaced by zeolite had about 19% higher compressive strength compared to the control specimen, and about 14% higher after 90 days of hydration [13]. Other authors declared that at 28 days the best compressive strength results (+7%) were obtained in specimens with 20% cement replaced by zeolite, and at 90 days, the best results (+13%) were obtained with 10% zeolite [14]. The authors, who investigated the influence of zeolite on the durability properties of concrete, found about 4% decrease in compressive strength at 28 days in the mixes where 15% of cement was replaced by zeolite in comparison to the control and a decrease of about 3% after 90 days of hydration. The 24-h water absorption tests showed a decrease of about 16% in the absorption rate compared to the control after 28 days, whereas the difference in the absorption rate after 90 days increased about 22% [15]. Another study on the durability of concrete revealed that at 28 days the compressive strength in the mixes with 10% cement replaced by zeolite decreased about 25% compared to the control, and after 90 days of hydration, this difference was 11%. The porosity tests showed that open porosity was about 18% higher when the binder contained 10% zeolite and increased with a higher zeolite content [16].

In summary, the research findings show that natural zeolites have pozzolanic properties. However, the replacement of part of the cement with zeolite is very likely to result in a decrease in compressive strength at 28 days due to late pozzolanic reactions. Moreover, there is a significant loss of compressive strength at 90 days and the loss in modified specimens is greater than the loss in the control specimen. Both positive and negative effects on open porosity are recorded, with the greatest positive effect of zeolite on the durability parameters, such as resistance to sulphate attack [17], alkali aggregate reaction [18], chloride diffusion [8], shrinkage strains and, consequently, crack reduction [19].

Soda lime glass, which is most widely used in the construction industry, contains high levels of amorphous SiO_2_ (≥65%), Na_2_O (≥12%) and CaO (≥5%). Such a chemical composition makes the glass powder an excellent pozzolanic material [20]. It should also be noted that the pozzolanic effect comes not only from the chemical composition, but also from other parameters, such as the type of glass, its particle size, the curing temperature and the free ions in the pore solution [21].

The reaction of ground glass in cement-based concrete starts in a similar way as the reaction of zeolite. It begins with the dissolution of CaO in the cement paste, which increases the pH of the pore solution. Then the glass powder starts dissolving in a hydrolysis reaction and thus supplies silica and sodium to a system already rich with Ca^2+^ [22]. A higher pH accelerates the dissolution of alkaline metals present in glass and SiO_2_ because silica fumes tend to dissolve faster in water at a pH between 12 and 13 [23]. A layer rich in silica forms on the surface of glass grains due to the incongruent dissolution of glass. The dissolved silica reacts with portlandite to form the outer C-S-H gel. Probably, the pozzolanic reaction of glass grains also produces two types of C-S-H, i.e., the inner hydration product formed within the original boundary of glass grains and the precipitated outer hydration product [24]. It means that glass powder, like zeolite, can form crystallisation centres and at the same time accelerate the hydration of tricalcium silicate.

According to the findings reported in scientific papers, the size of glass powder particles has a great effect on pozzolanic reactions. Some authors declare that the pozzolanic properties of glass become notable at particle sizes below approximately 300 μm and that glass powder particles of 100 μm in size can have a pozzolanic reactivity greater than that of fly ash at low cement replacement levels and after 90 days of curing [25]. Other authors report that glass powder could exhibit pozzolanic activity if finely ground to below 38 μm [26].

Although it is well known that a material containing high levels of Na_2_O and amorphous SiO_2_, such as glass, is a potential cause of ASR (Alkali Silica Reaction), there is research evidence showing that ASR potential highly depends on particle size. The high surface area of glass powder changes the kinetics of chemical reactions towards a beneficial pozzolanic reaction, where available alkali is utilised before the production of potential ASR gel [27]. Other researchers also found that glass powder, as a cement replacement material, demonstrates the potential to reduce deleterious expansion due to ASR reaction [28]. The same findings are stated by the authors who claim that partial replacement of cement by finely recycled waste glass can, in fact, reduce the ASR-related expansion [29].

The analysis of the data published in research papers shows both positive and negative effects on compressive strength caused by the replacement of part of the cement with ground glass. In their studies on the durability of mortars modified with ground glass, the authors report an approx. 12% drop in compressive strength at 28 days in specimens where 10% cement is replaced with ground glass, whereas the replacement of 20% of cement with glass results in a reduction of only about 3%. At 90 days, the reduction is 22% with 10% glass, and the strength of the specimens with 20% glass decreases 3% more than in the control specimens [30]. Other authors found that after 28 days the compressive strength of the specimens containing 5% of ground glass decreased approx. 5%, while after 90 days the specimens with 20% glass showed 13% higher strength than the control specimens. The open porosity of the specimens with 20% glass was about 11% lower at 90 days [31].

In summary, ground glass acts as a pozzolanic material and can improve the mechanical properties (compressive strength, flexural strength, etc.) of concrete. In addition to better mechanical characteristics, the microstructure [32] and durability-related properties of concrete, such as enhanced sorption, chloride permeability and freeze-thaw resistance [27], are improved due to the pozzolanic activity of glass powder.

The effect of the mixture of GP (glass powder) and other pozzolanic materials on the composition, structure and properties of cement concretes has not been widely researched yet. Research data are available on the effect of the mixtures of granulated slag, fly ash and RHA (rice husk ash) with ground glass. According to the studies on the combined effect of ground glass and RHA, the best results were obtained with the following binder ratios 85:10:5 and 85:7.5:7.5 (cement:GP:RHA), with the 28-day compressive strength of modified specimens about 13% lower than that of the control specimens, while at 90 days the difference was negligible [33]. The best composition found in another study with the same binder was 80:20:5. The authors report impressive results, as the achieved 28-day compressive strength was approx. 70% higher than in control specimens, and the 90-day strength was 98% higher. The total porosity studies show that after 28 days the total porosity of the mortar of the same composition was approx. 3.5% lower than the porosity of the control specimen, and after 90 days the difference was about 3.6% [34].

In this research, the combination of glass powder and natural zeolite in order to reduce capillary porosity of hardened cement paste and increase durability was investigated. The limitation of high hardened cement paste porosity after cement replacement with natural zeolite was tested. Additional modification with glass powder avoids an increase in water demand for equal cement mortar consistency and reduces hardened cement paste capillary porosity.

## 2. Materials and Methods

### 2.1. Materials

Portland cement CEM I 42,5 R complying with EN 197-1 requirements was used for the preparation of cement pastes and mortars. The amounts of minerals contained in the cement clinker (without gypsum) are: tricalcium silicate 57.80%, β-dicalcium silicate 22.15%, tricalcium aluminate 6.65% and tetracalcium aluminoferrite 13.40%.

The natural zeolite from the Transcarpathian region in western Ukraine is composed of clinoptilolite, heulandite and quartz. The amounts of oxides in cement, natural zeolite and glass powder are presented in Table 1. The main properties of the binder are given in Table 2.

The ternary diagram of the binders (Figure 1) shows the position of each binder in terms of the content of the main oxides and the position of the binder mixture that produced the best results of open porosity. The mixture of binders has shifted to the top of the diagram due to the change in composition and is positioned between Portland cement and granulated blast furnace slag.

Zeolite has a light brown colour, the particles are of irregular-angled shape, such as the particles of glass powder (Figure 2). Polycarboxylate polymer-based superplasticiser was used to control the consistency of the mortars. The superplasticiser had the following characteristics: active substance content 28%, pH 4.4 ± 1, density 1060 ± 20 kg·m^−3^. Sand fraction was 0/2, the fineness modulus of sand was 2.11. The sand was dried at 110 ± 5 °C before mixing. Cumulative particle size distribution of the mineral binders are presented in Figure 3.

### 2.2. Milling of Mineral Binders

The soda lime glass waste (from an insulated glass unit manufacturer) was ground using a planetary micro mill (capacity 1 t/h) with cylinder-shaped grinding bowls cased in stainless steel. Prior to grinding, the glass chippings were pre-crushed to 0/5 mm fraction.

The comminution takes place in the grinding bowl containing the glass particles and rotating around its own axis on the main disk, which rotates in the opposite direction. At a certain speed, the centrifugal force causes the ground glass and grinding balls to bounce off the inner wall of the grinding bowl, cross the bowl diagonally at an extremely high speed and impact the glass to be ground on the opposite wall of the bowl. Glass powder is delivered by compressed air to the separator where the particles of the required size are separated, and the remaining particles are returned to the mill where the grinding cycle is repeated until the required particle size is achieved.

The natural zeolite was milled in several stages. Initially, a drum mill with spherical steel grinding bowls was used, where the zeolite was milled to an average particle size of 59.3 μm in 48 h. Then the milling cycle of 48 h was repeated and the average particle size of 29.0 μm was obtained.

Since further milling was inefficient and did not have a significant effect on the particle size, it was decided to continue the milling using a wet process in a drum mill with ceramic milling bowls. The water to zeolite ratio was 2.35:1 and the milling cycle again lasted for 48 h. The obtained suspension had a milky consistency and had the following characteristics: a density of 1666 kg/m^3^, an absolute moisture content of about 211%, a water separation rate of about 10% of the total volume of the suspension over the period of two weeks and an average particle size of 3.6 μm.

### 2.3. Test Methods

Thermogravimetric and differential thermal analysis (registration of DTA curves) was conducted by the simultaneous thermobalance Linseis STA PT-1600. A corundum crucible with a sample of 50–60 mg was heated up to 1000 °C in an air atmosphere at a heating rate of 10 °C/min. The amount of calcium hydroxide formed was calculated by normalising the data according to the cement content in the mixtures, i.e., with the cement dilution effect evaluated.

Mineral composition analysis was done by DRON-7 (Bourevestnik JSC, St. Petersburg, Russia) X-ray diffraction meter with a 30 kV voltage, rotating Cu-Kα anode in the X-ray tube. The radiograms were done in 2θ at the interval of 5–60 degrees, by using an optical system adjusted to the Bragg–Brentano method, the sample rotating at 0.02 degrees and the detector rotating at 1 degree per minute. Crystallographica Search-Match v2.1 (Oxford Cryosystems, West Oxfordshire, UK) software and the database of crystal structures PDF-4+ (2019) was used for the analysis.

The calorimetric analysis was done with the calorimeter TAM AIR III and the data were analysed with Tam Air Assistant software. The temperature of the experiment was 25 ± 0.1 °C, and the water-to-solid ratio was 0.5 (measurement error < 0.03 W/g). The hydration process of the cement systems was measured for 60 h.

The open porosity of the cement pastes was measured using the modified method EN 1015-10 (determination of dry bulk density of hardened mortar). Wet specimens were weighed after the scheduled curing period (M_1_), with excess water wiped with a damp cloth from the surface of the specimens. Then the specimens were weighted in water after removing the air bubbles on the sides of the specimens (M_2_). Afterwards, the weighted specimens were dried in the drying oven at 60 ± 5 °C temperature until a constant mass was achieved. The specimen is considered to have achieved the constant mass if the difference in the weighting results after 24 h of drying does not exceed 0.2% of the dry specimen mass (M_3_). The capillary porosity (%) was calculated from the equation:(1)$$\mathrm{Porosity}=\left[\frac{M_{1}-M_{3}}{M_{1}-M_{2}}\right]\times 100$$

The problem of error in the results of M_1_, which is the mass of the saturated specimen in air, for small samples of the cement paste should be considered [35]. In order to increase the reliability of results, six specimens of each composition were tested and the obtained weighting results were processed using a Student’s t-distribution with the selected confidence level 95%.

The consistency of the cement pastes was determined by flow table method according to EN 1015-3. The density of the specimens was measured according to EN 1015-6. The flexural and compressive strength was tested according to EN 1015-11. The mortar prisms (40 × 40 × 160 mm) were initially tested by loading in three points to failure and then compressed by placing between two bearing plates of 40 × 40 mm. The machine has two steel supporting rollers spaced 100 mm apart, and the third steel roller of the same length and diameter located centrally between the supporting rollers. The load was applied at a uniform rate of 50 N/s. The mortars were compressed at a uniform rate of 200 N/s.

The porosity of fresh mortars was calculated by evaluating the relationship between the determined density of the mix and the maximum theoretical density, taking into account the quantities and specific densities of all components of the mix.

### 2.4. Mix Proportioning and Sample Preparation

The ratios of cement to mineral binders in cement-based mortar mixtures were converted according to the densities of the materials used: water to binder ratio was 0.5; sand to binder ratio was 3:1; superplasticiser was added at 0.8% by weight of cement. Cement mortars were mixed with the composition presented in Table 3 and Table 4.

Mortars were mixed according to the standard method prescribed by EN 480-1 Part 1: Reference concrete and reference mortar for testing. Specimens with the dimensions 40 × 40 × 160 mm were formed in metal moulds. The mortar was poured in two layers and compacted on the vibration table. The prisms were conditioned for 24 h in the forms at 20 ± 2 °C, demoulded afterwards and further cured in water at the same temperature.

The cement pastes were formed using the same cement/mineral binder ratio as the ratio used in the mortars with the water/binder ratio 0.4. The paste was mixed according to the standard method prescribed by the standard EN 196-7 Part 7: Methods of taking and preparing samples of cement. The specimens were formed in tabled-shaped PVC moulds with a diameter of Ø 60 mm and a height of 25 mm. The prisms were conditioned for 24 h in the moulds at 20 ± 2 °C, demoulded afterwards and further cured in water at the same temperature. The composition of the cement paste samples is presented in Table 5.

Microcalorimetry tests were conducted with the mixtures containing 85% of cement and 15% of natural zeolite with different average particle size and a mixture containing 90% of cement and 10% of milled glass. The water/binder ratio was 0.5 in all mixtures.

## 3. Results and Discussion

### 3.1. Workability and Porosity of Fresh Mortars

The results provided in Table 6 show that zeolite and glass powder used to replace cement have different effects on the workability of the mortars. Zeolite added at 10% increases the mortar slump flow, which starts decreasing at a content of zeolite higher than 10%. In the case of glass powder, the mortar slump flow increases irrespective of the glass powder content. The mortar slump flow of the mixtures containing zeolite is mainly influenced by the porous structure of the zeolite and high specific surface area of zeolite particles, resulting in high absorption rate and thus reduced amount of free water [36]. The angular shape of zeolite particles may increase the friction force between the particles [37]. Glass, in contrast to zeolite, has a low absorption capacity, irrespective of the particle surface area and particle shape.

Therefore, it can increase the workability of the mix, which is confirmed by other authors [38]. It should be noted that there is a relationship between particle size and the amount of glass powder. Authors report [39] that the slump flow did not decrease in the mix containing up to 20% of glass powder with an average particle size of 45 μm. However, the workability may decrease when the glass particles are smaller than 10 μm.

When a portion of cement is replaced with a mix of zeolite and glass powder (Table 7) with the binder ratio 75:10:15, the mortar slump flow is always higher than that of the control specimen and does not differ. When the binder ratio (cement:zeolite:glass powder) in the mix is changed to 75:15:10, the slump flow depends on zeolite particle size, because smaller particles of zeolite reduce the slump flow of the mix. Nevertheless, the mortar slump flow in all cases is higher than the slump of the control specimen.

The porosity of the mix is increased by replacing part of the cement with zeolite, but the porosity of the fresh mix is not affected by increased zeolite content. The porosity increases slightly when part of the cement is replaced with glass powder, whereas the increased glass powder content does not have a significant effect on porosity, as in the mixtures with zeolite. When part of the cement is replaced with a mixture of minerals, the porosity increases significantly, irrespective of the zeolite particle size or the proportion of binders, but it should be noted that the porosity is the lowest in the mixture made with a zeolite suspension and a binder ratio of 75:15:10. Since zeolite has a high absorption capacity and has been found to be able to absorb about 34% of water, presumably the 14.2% decrease in porosity was caused by using the zeolite suspension instead of dry zeolite in the mix, and the improvement in porosity parameters might have been caused by the displacement of air in zeolite voids by the water.

### 3.2. Hydration Progress and Heat Release

First of all, it should be noted that due to the wetting of the particle surface, the exothermic peak is significantly lower (Figure 4a) in the mixtures where part of the cement is replaced with glass powder, than in the mixtures of other compositions. The most likely reason is almost zero absorbency of the glass.

However, the induction period in the mixtures with glass powder is the shortest. After two hours of hydration, the heat release in the specimen containing glass powder is comparable to the specimens where part of the cement is replaced with zeolite that has an average particle size of 59.3 or 29.0 μm. After about 6 h of hydration, the heat flow (Figure 4b) in the specimen containing only cement exceeded the heat flow of the specimen where part of the cement was replaced with glass powder, but the heat flow in the specimen containing glass powder became the highest between 22 and 32 h of hydration.

The comparison of the samples with different zeolite particle sizes showed that the maximum heat flow was recorded in the specimen containing zeolite with a particle size of 3.6 μm. The induction period was the same as in other specimens (except for the specimens containing glass powder), showing the maximum heat flow for about 6 h and then decreasing to the lowest value compared to the other specimens. The total amount of heat (Figure 5) released until 11 h of hydration was the highest in the specimen containing zeolite with 3.6 μm particles. After 11 h of hydration, the highest amount of heat was released in the control mixture with 100% of cement. The results show that after 11 h the acceleration was insufficient to compensate for the dilution effect [37]. On the one hand, the obtained results confirm the results of other authors indicating that, during the first hours of hydration, zeolite has a direct effect on the hydration of cement clinkers due to the surface wetting effect [40], which is attributed to the high absorption capacity and the specific surface area of zeolite particles, and an indirect effect caused by zeolite particles acting as crystallisation centres, leading to accelerated hydration of C_3_S [41]. On the other hand, there are studies proving that the zeolite admixture accelerates the initial and final setting time of cement pastes [7] and these findings confirm the previous statements regarding the acceleration of C_3_S hydration.

### 3.3. Physical and Mechanical Properties of the Mortars

The mechanical properties of mortars No 1 were tested in order to find the optimum amount of the mineral material that could be used to achieve the best zeolite and glass powder ratio. Our previous tests [42] showed that the pozzolanic activity of the zeolite used in our research became evident after 60–90 days. Therefore, a decision was made to conduct tests only after 90 days. Figure 6a shows that zeolite has a negative effect on the flexural strength, i.e., all specimens where part of the cement was replaced with zeolite demonstrated lower resistance to bending than the control specimens, whereas the specimens where cement was replaced with glass power resisted failure in bending, in a similar way to the control specimens, ranging within a ±2.3% limit.

The results of compressive strength tests show that the strength of all specimens where part of the cement was replaced with glass powder was higher than the strength of control specimens. The highest effect (Figure 6b) was achieved in the specimens containing 5% and 10% glass. The tests of the specimens containing zeolite show that 10% and 15% zeolite had a minimum (up to 1.5%) positive effect, whereas the results of the specimens containing 5% and 20% zeolite demonstrate 12.0% and 8.3% lower results, respectively. The best compressive strength result (approx. 5% higher than the strength of the control specimen) after 90 days of curing was achieved in the specimen containing 10% glass powder.

Authors also report [38] that the 10% level of cement replacement with glass powder gave the biggest compressive strength in the mortar tested. Other authors found that at 60 days of curing a larger amount of outer C-S-H was formed around glass grains in the cement paste prepared with fine glass grains. In this way, a reaction rim is created that can be seen around glass grains where the spaces between grains are well filled [24].

The evaluation of the results reported above led to the conclusions that the highest relative effect was achieved when 10% and 15% of cement was replaced with one of the mineral materials. Therefore, compositions made with 75% cement and a mixture of mineral materials added at the ratios of 15:10 and 10:15 and using zeolite of three different average particle sizes were further tested for mechanical properties. The results of these tests are presented in Figure 7. The results of compressive strength after 28 days given in Figure 7a show that pozzolanic reactions in all specimens are not sufficient to compensate for the dilution effect, which is the cause of the strength loss from 22% to 32%. The main difference observed was 5% better results obtained with zeolite of smaller average particle size. The results after 90 days are given in Figure 7b and show that the strength difference between the test specimens and control specimens became smaller, i.e., from 19% to 30%, while the effect of particle size remained the same. The best composition of all specimens tested was with the binder ratio 75:15:10 with an average zeolite particle size of 3.6 μm.

The decrease in compressive strength compared to the control specimen can be explained by the slow pozzolanic reaction. As some zeolite particles did not react with Ca(OH)_2_, the degree of reaction at 90 days of curing was not sufficient to compensate for the decrease in the cement content. This hypothesis is supported by the peaks of one of the zeolite minerals, namely heulandite, at its characteristic 2 theta angles in the XRD intensity plot (Figure 8). 

The calculations of Ca(OH)_2_ content (Table 8) showing that the mortars, in which part of cement was replaced by a mixture of minerals, have a higher Ca(OH)_2_ content than the control specimens confirm the previous statements regarding the slow pozzolanic reaction. A 12.1% decrease in Ca(OH)_2_ was observed only in the mortar where the zeolite with the smallest particle size was used. According to other authors, at 28 days, the Ca(OH)_2_ content in cements containing 10–20% of zeolite was 14–23% lower than in the pastes with plain cement and 19–34% lower at 360 days [43]. This negative effect may have been caused by the replacement of 25% of cement with a mixture of zeolite and ground glass, which reduced the overall pH of the system due to the dilution effect of the cement, which influenced the dynamics of the dissolution of zeolite crystals, followed by the crystallisation of hydration products on the surface of zeolite particles and the transition to a significantly slower diffusion process.

### 3.4. Porosity by Immersion

The tests of water absorption by immersion and open porosity (Figure 9) showed that, in contrast to the results of mechanical strength, water absorption of the specimens of all compositions where part of the cement was replaced with mineral material was higher than the absorption of control specimens after 28 and 90 days of curing. After 28 days of curing, the open porosity in the specimens containing zeolite increased with a higher content of cement replaced. A similar trend was observed in the specimens containing glass powder.

However, it should be noted that a small decrease in the open porosity was observed in the specimens containing 15% glass powder. Although after 90 days of curing the porosity difference decreased, the modified specimens demonstrated higher open porosity results than control specimens. After 90 days of curing, a slight decrease in porosity values was again observed in the specimens containing 15% glass powder.

The porosity tests of mortars where 25% cement was replaced by a mixture of mineral additives at different proportions and with different zeolite particle sizes showed (Figure 10) that, after 28 days of curing, the open porosity in all modified specimens was higher than in the control specimen. The same tests done after 90 days of curing showed that the composition where cement was replaced with a mixture of 15% zeolite with a particle size of 3.6 μm and 10% soda lime glass had the lowest open porosity. It should also be noted that density and porosity tests of fresh mixtures revealed that the composition described above had the highest density and the lowest porosity, which was 10% higher than the porosity of the control specimen. These results indicate a probability that these compositions may have high closed porosity and, consequently, high frost resistance.

### 3.5. X-ray Diffraction Analysis

The X-ray diffraction analysis revealed the absence of ettringite in all compositions where part of the cement was replaced by a mixture of mineral additives. Presumably, the ettringite content reduced in proportion to the reduced cement content in the mixture and the remaining amount of ettringite was not sufficient to be captured by this method of analysis.

These compositions contained free heulandite, as well as calcium aluminosilicate hydrates or oxides that were not found in the control composition. The main difference when comparing the mixtures with different zeolite particle sizes is that only the mixture with 3.6 μm zeolite particles has peaks of kamaishilite, which are absent in the compositions with larger particles. Presumably, the formation of this mineral could be one of the reasons for the decrease in open porosity.

### 3.6. Thermogravimetric Analysis

The results of thermogravimetric analysis converted to the mass of cement showed a significant reduction in calcium hydroxide (12.1%) in the specimen where zeolite with the smallest particle size was used. No pozzolanic reactions were observed in other compositions, as they contained higher levels of calcium hydroxide than the control specimen. These results correlate well with the open porosity test results, which show that the specimen with the smallest zeolite particle size had a lower open porosity than the control specimen and that other specimens with zeolite particle sizes of 29.0 and 59.3 μm had a higher open porosity than the control specimen. The same is confirmed by the results of the mechanical properties tests, which show that the specimens with the smallest zeolite particle size had the best compressive strength.

The results of the test of mechanical properties are similar, showing that the specimens with the smallest zeolite particle size had the highest compressive strength.

However, these specimens had a cement dilution effect and the pozzolanic reaction was not sufficient to compensate for the reduction of cement content. Therefore, all modified specimens had lower strength results than the control specimen.

Minor endothermic effects at approx. 190 °C and 230 °C temperature marked by dotted lines in Figure 11 should also be noted. As the control specimen did not have such effects, it is possible that they can be caused by calcium aluminate hydrates, which were observed in the XRD images. Taking into consideration the decomposition temperatures of typical cement clinker hydration products described in the literature [44], it is very likely that calcium aluminate and calcium aluminate silicate hydrates decompose at these temperatures.

## 4. Conclusions

Natural zeolite and soda lime glass have pozzolanic properties, which become apparent in cement concretes after 60 to 90 days of curing, but they have different effects on the early hydration of Portland cement according to microcalorimetry results. Glass powder has no exothermic effect due to the wetting of the particle surface, although the dynamics of the first peak of the heat flow diagram is very close to that of zeolite. The total heat released (J/g) by a cement mixture with glass powder is significantly lower than that of cement, or of cement modified with zeolite.Soda lime glass slightly increases the flexural strength of the mortar when added at 10% and 15% by weight of cement and it has the greatest effect on the compressive strength when added at 5% or 10% by weight of cement. Meanwhile, natural zeolite has a negative effect on the flexural strength, but it increases the compressive strength when added at 15% by weight cement. When cement is replaced with 10% zeolite, there is only a slight increase in the compressive strength.Both natural zeolite and soda lime glass mineral additives have a negative effect on the open porosity of hardened cement paste, i.e., the increase in the proportion of one of the additives at the expense of cement results in increased water absorption, which can have a negative effect on the durability of cement-based composites. The open porosity (water absorption) of hardened cement mortar is directly related with water demand for equal consistency of cement mortar.The mixture of mineral additives has the highest effect in terms of open porosity and mechanical resistance to failure when added at the proportion 75:15:10 and when zeolite with an average particle size of about 3.6 μm is used. This effect could be explained by the presence of calcium silicate aluminate hydrates (minerals identified as katoite and kamaishilite) in the cement matrix after hydration. Presumably, these minerals fill some of the pores formed during hydration, but the effect of these minerals is not sufficient to compensate for the cement dilution effect, which leads to the reduction in mechanical strength.The size of zeolite particles has a significant effect both on the fresh and hardened cement–concrete. The results of workability tests show that the mixture with the smallest zeolite particle size had the lowest mortar slump flow, as well as the lowest porosity and the highest compressive strength. Microcalorimetry results show that the exothermic effect is the highest both in terms of heat flow and the heat released. The results of thermogravimetric analysis show that the specimens with the finest zeolite particles had the highest pozzolanic effect in terms of Ca(OH)_2_ mass loss per unit of mass of cement, which explains the better strength results and, to some extent, lower open porosity.

## Figures and Tables

**Figure 1 materials-15-04219-f001:**
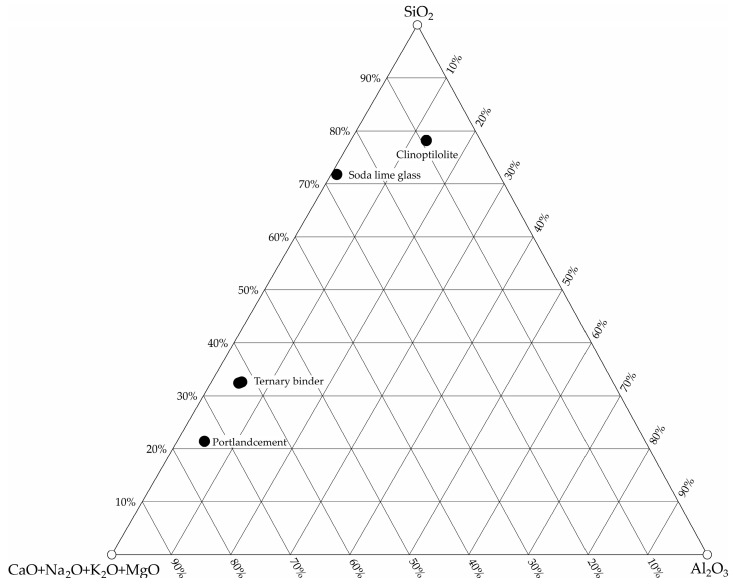
Ternary diagram of the binders.

**Figure 2 materials-15-04219-f002:**
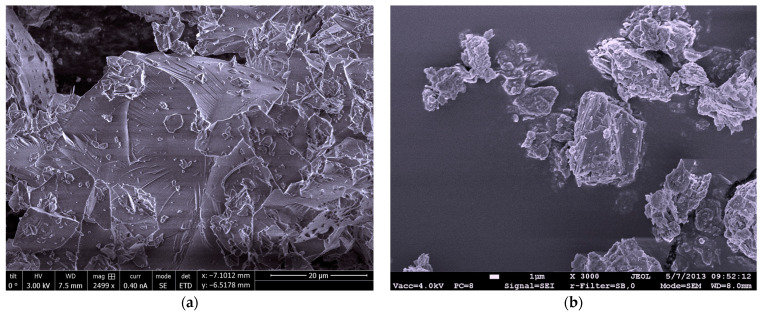
SEM images of binder particles: (**a**) glass powder (**b**) natural zeolite (clinoptilolite).

**Figure 3 materials-15-04219-f003:**
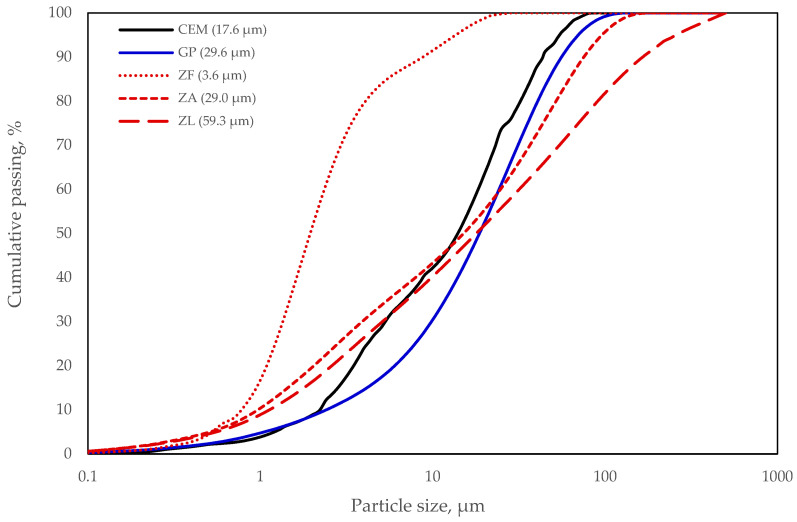
Cumulative passing chart of binder particle size.

**Figure 4 materials-15-04219-f004:**
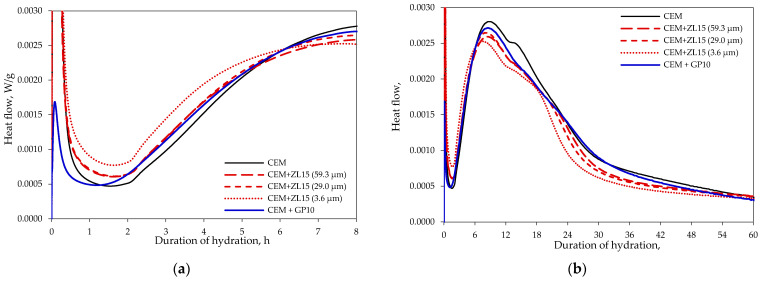
The dynamics of heat release: (**a**) 8 h of hydration; (**b**) 60 h of hydration.

**Figure 5 materials-15-04219-f005:**
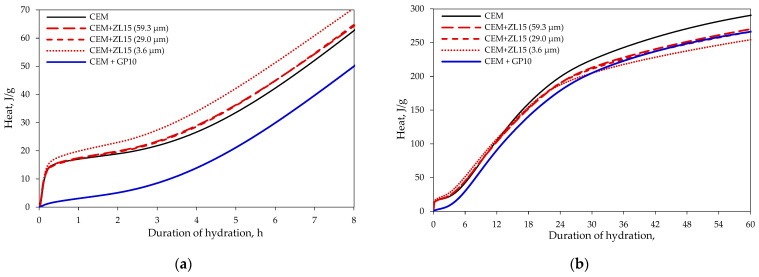
The amount of released heat: (**a**) 8 h of hydration; (**b**) 60 h of hydration.

**Figure 6 materials-15-04219-f006:**
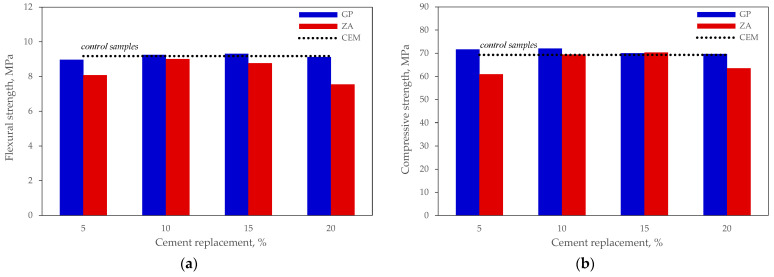
Results of the mechanical tests of mortars No 1 after 90 days: (**a**) flexural strength; (**b**) compressive strength.

**Figure 7 materials-15-04219-f007:**
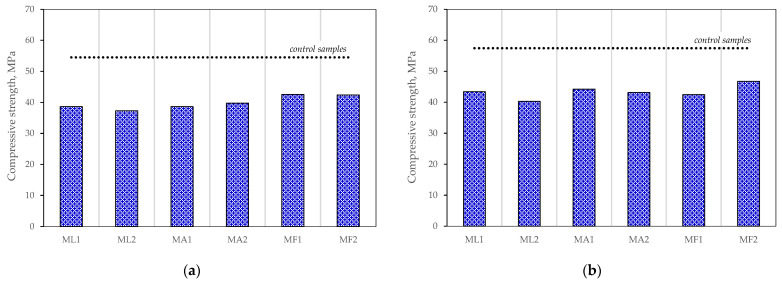
Results of the mechanical properties tests of mortars No 2: (**a**) after 28 days; (**b**) after 90 days.

**Figure 8 materials-15-04219-f008:**
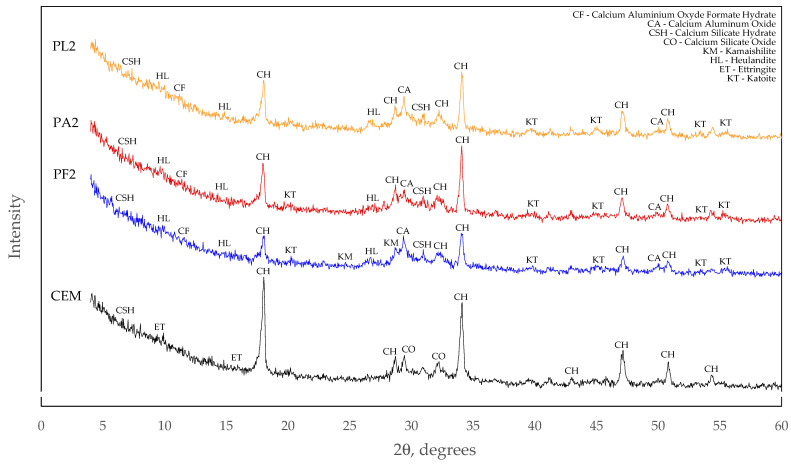
X-ray diffraction images of hardened cement pastes at 90 days.

**Figure 9 materials-15-04219-f009:**
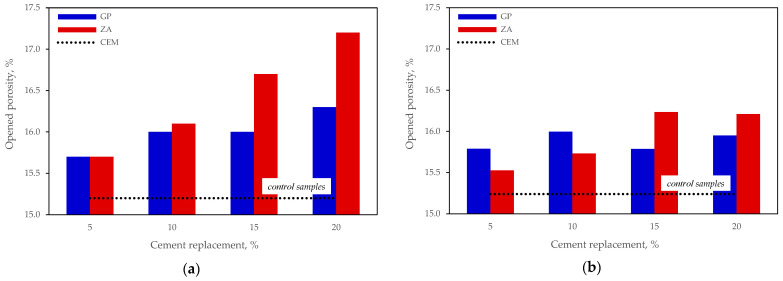
Open porosity of mortars No 1: (**a**) after 28 days; (**b**) after 90 days.

**Figure 10 materials-15-04219-f010:**
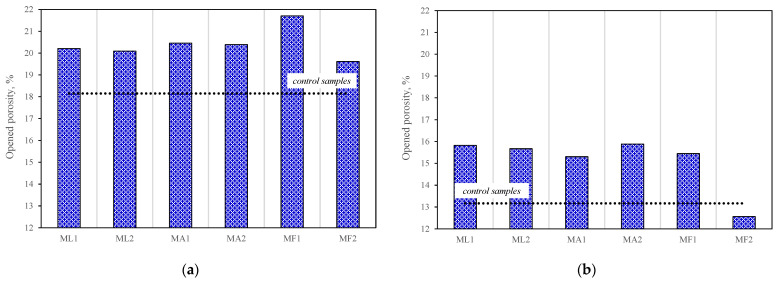
Open porosity of mortars No 2: (**a**) after 28 days; (**b**) after 90 days.

**Figure 11 materials-15-04219-f011:**
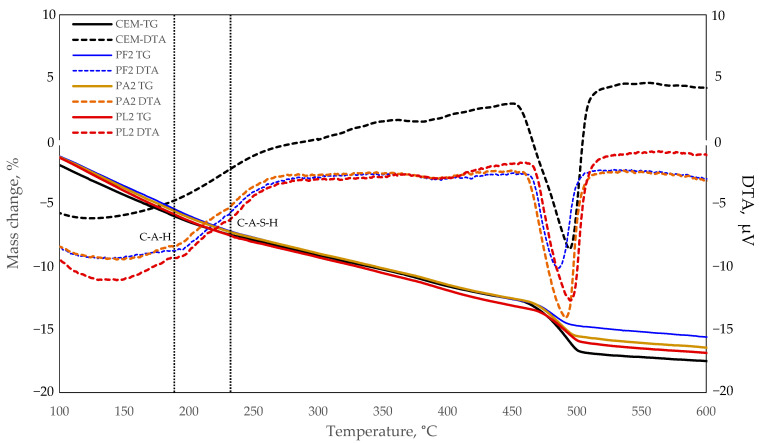
Graphs of thermogravimetric tests on hardened cement paste.

**Table 1 materials-15-04219-t001:** Chemical compositions of cement, natural zeolite and soda lime glass.

Material	Oxide Content (wt.%)								
	CaO	SiO_2_	Al_2_O_3_	Fe_2_O_3_	MgO	SO_3_	K_2_O	Na_2_O	LOI
Cement	61.4	19.5	5.0	3.1	3.9	2.5	1.1	0.1	3.4
Zeolite	3.3	72.5	12.5	1.7	0.6	-	3.6	0.2	5.6
Glass	10.1	71.4	1.0	0.1	3.9	0.3	0.3	12.8	0.2

**Table 2 materials-15-04219-t002:** Properties of cement, natural zeolite and glass.

Material	Abbreviation	Specific Gravity (kg/m^3^)	Mean Particle Size (μm)	Specific Surface by Blaine (m^2^/kg)
Cement	CEM	3150	17.6	440
Zeolite	ZL	2350	59.3	320
	ZA		29.0	760
	ZF		3.6	3800
Glass powder	GP	2570	29.6	335

**Table 3 materials-15-04219-t003:** Compositions of cement mortars No 1 (for the mixture volume of 1 m^3^).

Materials	Mixture Designations								
	CEM	CZ1	CZ2	CZ3	CZ4	CG1	CG2	CG3	CG4
Cement (kg)	512.9	487.3	461.6	436.0	410.4	487.3	461.6	436.0	410.4
Zeolite (%)	-	5	10	15	20	-	-	-	-
Zeolite (kg)	-	19.3	38.6	57.9	77.2	-	-	-	-
Glass powder (%)	-	-	-	-	-	5	10	15	20
Glass powder (kg)	-	-	-	-	-	20.9	41.9	62.8	83.8
Water (kg)	256.5								
Sand (kg)	1538.8								
Superplasticiser (kg)	4.1								

**Table 4 materials-15-04219-t004:** Compositions of cement mortars No 2 (for the mixture volume of 1 m^3^).

Materials	Mixture Designations						
	CEM	ML1	ML2	MA1	MA2	MF1	MF2
Cement (kg)	512.9	384.7	384.7	384.7	384.7	384.7	384.7
Zeolite particles size (μm)	-	59.3		29.0		3.6	
Zeolite (%)	-	10	15	10	15	10	15
Zeolite (kg)	-	38.6	57.9	38.6	57.9	38.6	57.9
Glass powder (%)	-	15	10	15	10	15	10
Glass powder (kg)	-	62.8	41.9	62.8	41.9	62.8	41.9
Water (kg)	256.5						
Sand (kg)	1538.8						
Superplasticiser (kg)	4.1						

**Table 5 materials-15-04219-t005:** Cement paste compositions.

Materials	Mixture Designations						
	CEM	PL1	PL2	PA1	PA2	PF1	PF2
Water/binder ratio	0.5						
Cement (%)	100	75		75		75	
Zeolite particles size (μm)	-	59.3		29.0		3.6	
Zeolite (%)	-	10	15	10	15	10	15
Glass powder (%)	-	15	10	15	10	15	10

**Table 6 materials-15-04219-t006:** The main properties of fresh cement mortars No 1.

Property	Mixture Designations								
	CEM	CZ1	CZ2	CZ3	CZ4	CG1	CG2	CG3	CG4
Flowability, mm	150	165	170	160	155	155	170	165	170
Density, kg/m^3^	2233	2153	2142	2145	2137	2219	2219	2201	2185
Porosity, %	2.33	5.93	6.16	5.71	5.83	2.83	2.64	3.27	3.79

**Table 7 materials-15-04219-t007:** The main properties of fresh cement mortars No 2.

Property	Mixture Designations						
	CEM	ML1	ML2	MA1	MA2	MF1	MF2
Flowability, mm	180	210	210	210	205	210	180
Density, kg/m^3^	2229	1964	1923	1953	1935	1929	1996
Porosity, %	3.56	16.2	18.5	16.8	17.8	18.3	14.2

**Table 8 materials-15-04219-t008:** Results of the calculation of calcium hydroxide content per unit of mass of cement.

Composition	M_CH_, mg/mg	Difference, %
Control	0.1498	-
PL2	0.1876	20.1
PA2	0.2004	25.3
PF2	0.1336	−12.1

## Data Availability

Data are contained within the article.

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
