# Peer review of "Use of Natural Zeolite and Glass Powder Mixture as Partial Replacement of Portland Cement: The Effect on Hydration, Properties and Porosity"

_materials, 2022, doi:10.3390/ma15124219_

Round 1

Reviewer 1 Report

 Article 1

Use of Natural Zeolite and Glass Powder Mixture as Partial 2 Replacement of Portland Cement: The Effect on Hydration, 3 Properties and Porosity

Authors: Dalius Kriptavičius 1,*, Giedrius Girskas 1 and Gintautas Skripkiūnas

The paper deal with the use of two different mineral admixtures separately and in combination. The study is interesting, well presented and written but there are several weaknesses and comments that have to be addressed before this paper can be accepted for publication in this Journal. Here are some general comments but more detailed comments are embedded in the annotated pdf file.

There is too much reporting and describing and very little, in deep, discussion of the results. The authors are requested to significantly improve their discussion and results analysis.

There is an issue with numbering Figures!!! This has made the review very hard!!!

Please revise it.

It seems that your zeolite did not play either a physical role (as a filler material, filling the pores) nor a chemical one (pozzolanic reaction) as it had 72.5% silica. So, for what purpose your zeolite was incorporated???

The effect of the size of zeolite on the mortar`s properties was not clearly identified.

Conclusions should be rewritten in a summarized way. Better in a bullet point format.

Author Response

Dear Reviewer,

First of all, I would like to apologize for the inconvenience of reading our article, as it was converted to an older version of Word at the last minute and the format has changed as a result.

In attached file you may see the responses to your comments.

Thank you very much for your time, valuable comments and suggestions.

Reviewer 2 Report

This manuscript is about the use of natural zeolite and glass powder mixture as partial replacement of Portland cement: the effect on hydration, properties, and porosity

This manuscript needs some improvements.

Abstract:

Add something about the mechanical performance/ results of the research. 

Add something about the benefits results of the research. 

Introduction

-Well written.

Materials and Methods:

-A comprehensive research framework missing- to follow the research is steps are missing. Add framework-flowchart and write this section in stepwise pattern.

- add detail about testing procedure of mechanical properties…. Compressive, flexural strength testing procedure missing? Add heading 2.5

Results:

-If we see section 3.3. Physical and mechanical properties of the mortars…. Fig.5.shows that GP is a potential replacement of cement….

-However, Glass powder (GP) is already an established binder and replacement of cement. So, what new has been done in this research.??? Main issue

-According to the Fig.6…again GP and NZ are also not competitors of cement…again failure….

-If we are observing failure in mechanical properties… hydration and porosity doesn’t matter a lot.

-I think in this section if authors add comparative analysis of results with previously published papers (4-5 reference results-bar graphs) (i.e., comparison with compressive, flexural performance etc.)..We may explore some different trends….suggestion.

Please compare the results of this research with existing published research with existing materials. Line or bar charts can be added. [Please add]

Discussion:

This section must contain implications for research, practice and/or Field: Does the paper identify clearly any implications for research, practice and/or society? Does the paper bridge the gap between theory and practice? How can the research be used in practice (economic and commercial impact), to influence technical policy, in research (contributing to the body of knowledge)? Add something for field professionals. [Please add]

Limitations of the study:

Please add as heading about the limitations of the study.

Also see heading 4 is missing. Check heading numbers

Author Response

Dear Reviewer,

First of all, I would like to apologize for the inconvenience of reading our article, as it was converted to an older version of Word at the last minute and the format has changed as a result.

According your comments and suggestions:

Add something about the mechanical performance/ results of the research. 

Add something about the benefits results of the research. 

Added to abstract

Add detail about testing procedure of mechanical properties…. Compressive, flexural strength testing procedure missing? Add heading 2.5

Detailed testing procedure of mechanical properties added.

If we see section 3.3. Physical and mechanical properties of the mortars…. Fig.5.shows that GP is a potential replacement of cement….

However, Glass powder (GP) is already an established binder and replacement of cement. So, what new has been done in this research.??? Main issue

This study investigates the complex effect of zeolite and glass powder mixture on its porosity parameters. The data presented in Figure 7 show that using these materials separately and replacing cement with up to 20% zeolite or glass increases the open porosity, while replacing 25% cement with a mixture of these minerals has achieved a slight but positive improvement in porosity parameters.

According to the Fig.6…again GP and NZ are also not competitors of cement…again failure….

If we are observing failure in mechanical properties… hydration and porosity doesn’t matter a lot.

As mentioned earlier, strength is not the ultimate goal although we already have ideas on how to increase strength to control. Environmental resistance is important like strength. In aggressive environmental conditions with a lot of freezing-thawing cycles at high humidity (e.g. road construction, sea water impact and others) durability parameters of concrete are crucial.

I think in this section if authors add comparative analysis of results with previously published papers (4-5 reference results-bar graphs) (i.e., comparison with compressive, flexural performance etc.)..We may explore some different trends….suggestion.

Please compare the results of this research with existing published research with existing materials. Line or bar charts can be added. [Please add]

As the effect of the mixture of these minerals admixture on the properties of cement concrete has not been studied (at least there are no publications on this topic) we do not have the opportunity to make a comparison with the results of other authors. We supplemented results and discussion section with a comparison with the results obtained by other authors when part of the cement is replaced by zeolite or glass powder.

This section must contain implications for research, practice and/or Field: Does the paper identify clearly any implications for research, practice and/or society? Does the paper bridge the gap between theory and practice? How can the research be used in practice (economic and commercial impact), to influence technical policy, in research (contributing to the body of knowledge)? Add something for field professionals. [Please add]

The final target of our study is to increase the resistance of cement based composites to sea water, as concrete is exposed to different natures in this field of application. The effects of chlorides, sulphates, acids, freeze-thaw cycles are being considered, and the possibility of increasing the resistance to this aggressive environment by replacing part of the cement with a mixture of mineral additives is being considered. Porosity parameters are very important in preventing the above-mentioned chemical compounds from entering the concrete structure, as well as the chemical resistance of cement minerals. Of course, the porosity studies presented in the article do not provide these properties and the further analysis required is accompanied by appropriate tests. Natural zeolite does not requires for some special technological process only mechanical treatment. Waste glass as a secondary raw material can only be used in a relatively narrow field and can be efficiently utilized in a complex way with natural pozzolana.

Also see heading 4 is missing. Check heading numbers

Changed

Thank you for your time and valuable comments.

Reviewer 3 Report

This is interesting and timely work. Some comments:

  1. What is the innovation of this study? according to my knowledge, there are plenty of works investigating the waste replacement of cement concrete.
  2. Line 63: the authors only mentioned the decomposition mechanism of zeolites, how does the 15% come from?
  3. The literature review is quite poor, please include the previous studies on using glass powder and pozzolanic materials in cement materials. The advantages, drawbacks, and max. contents of waste materials should be highlighted.
  4. A summary section should be included by the end of the introduction.
  5. Figure 1 is a little bit blurry, please the content in the plot if possible.
  6. Table 5: how do the authors decide the content ratios?
  7. Chapter 3.3: please use the same color for the same material.
  8. The overall analysis is reliable, however, this paper is more like a technical report instead of a scientific paper.
  9. The authors mentioned in the introduction that the purposes of using waste materials in cement concrete are resource efficiency and reduced emissions. How about the emission reduction with such a 25% replacement?

Author Response

Dear Reviewer,

First of all, I would like to apologize for the inconvenience of reading our article, as it was converted to an older version of Word at the last minute and the format has changed as a result.

According your comments and suggestions:

1. What is the innovation of this study? according to my knowledge, there are plenty of works investigating the waste replacement of cement concrete.

You’re right, with zeolite or ground glass there’s actually a lot of published research, but our goal was to investigate the effect of a complex admixture on the porosity parameters of cement based composites, to increase resistance to aggressive environments.

2. Line 63: the authors only mentioned the decomposition mechanism of zeolites, how does the 15% come from?

15% comes from our previous investigation, literature analysis and test results from mortars No 1 in this investigation.

3. The literature review is quite poor, please include the previous studies on using glass powder and pozzolanic materials in cement materials. The advantages, drawbacks, and max. contents of waste materials should be highlighted.

We supplemented the introduction with data from articles in which part of the cement is replaced by zeolite or ground glass and presented the strength results, optimal amount, and particle size.

4. A summary section should be included by the end of the introduction.

We supplemented the introduction with a summary

5. Figure 1 is a little bit blurry, please the content in the plot if possible.

Changed to better quality picture

6. Table 5: how do the authors decide the content ratios?

In this study, we found that the maximum strength is achieved when zeolite is used at 15% or glass at 10%, and we also evaluated the effect of the additive on the flow to have mixtures of similar consistency.

7. Chapter 3.3: please use the same color for the same material.

Changed

8. The overall analysis is reliable, however, this paper is more like a technical report instead of a scientific paper.

Sometimes it is difficult to find a balance between theory and practice. Because there are very few studies using a complex additive made of natural pozzolans and glass, we have done a lot of researches and reduced the analytical portion accordingly.

9. The authors mentioned in the introduction that the purposes of using waste materials in cement concrete are resource efficiency and reduced emissions. How about the emission reduction with such a 25% replacement?

We do not have data on how much energy we can reduce, but we believe that the producing of cement clinkers requires more energy, and on the other hand, increasing the durability of cement based composites should also reduce its need in terms of a longer material life cycle.

Thank you for your time and valuable comments.

Reviewer 4 Report

This paper studies the effects of natural zeolite (clinoptilo-lite) and soda lime glass powder mixture on hydration, mechanical properties and porosity of concrete. The microstructure of hardened cement paste modified with the aforementioned mineral additions was analyzed by means of mechanical, microcalorimetry, X-ray diffraction tests and thermogravimetric analysis. In summary, the research is interesting and provides valuable results, but the current document has several weaknesses that must be strengthened in order to obtain a documentary result that is equal to the value of the publication.

1)The abstract is complete and well-structured and explains the contents of the document very well. Nonetheless, the strength improvement and porosity of cement mortar under the optimum dosage can be briefly introduced in this part. 
2)In the introduction, the author introduces the possibility of substituting natural zeolite and sodium-calcium glass powder for cement from the perspective of high energy consumption. However, the author introduces the action mechanism of the two admixtures in the hydration process of cement at length, but does not show the improvement of the properties of cement-based materials by their actions alone or together in previous experiments.
3)The first paragraph introducing the research topic may present a much broad and comprehensive view of the problems related to your topic with citations to authority references (Combined effects of nano-silica and silica fume on the mechanical behavior of recycled aggregate concrete; Compressive properties of rubber-modified recycled aggregate concrete subjected to elevated temperatures). 
4)In Fig.4 and Fig.5, error line of figure and description occurs. 
5)What are the main mineral components of the high peaks between 5° and 15° in the XRD figure? The mineral components other than Heulandite need to be explained.
6)In the conclusion part, the optimum dosage of natural zeolite and sodium-calcium glass powder mixture should be put forward considering the strength, workability and hydration reaction of cement mortar. 
7)It should mention the scope for further research as well as the implications of the study. I recommend including the limitations regarding the consideration of damage indicated in this review in the limitations assessment. This part of the document can be improved and completed with more rigour. The advantages of the use of this mixture to the performance of cement mortar should be mentioned briefly. 

Author Response

Dear Reviewer,

First of all, I would like to apologize for the inconvenience of reading our article, as it was converted to an older version of Word at the last minute and the format has changed as a result.

According your comments and suggestions:

1) The abstract is complete and well-structured and explains the contents of the document very well. Nonetheless, the strength improvement and porosity of cement mortar under the optimum dosage can be briefly introduced in this part.

Corrected

2) In the introduction, the author introduces the possibility of substituting natural zeolite and sodium-calcium glass powder for cement from the perspective of high energy consumption. However, the author introduces the action mechanism of the two admixtures in the hydration process of cement at length, but does not show the improvement of the properties of cement-based materials by their actions alone or together in previous experiments.

Comparisons with the results obtained by other authors when using such pozzolanic materials are detailed in the section.

3) The first paragraph introducing the research topic may present a much broad and comprehensive view of the problems related to your topic with citations to authority references (Combined effects of nano-silica and silica fume on the mechanical behavior of recycled aggregate concrete; Compressive properties of rubber-modified recycled aggregate concrete subjected to elevated temperatures). 

Added to literature analysis.

4) In Fig.4 and Fig.5, error line of figure and description occurs. 

Corrected

5) What are the main mineral components of the high peaks between 5° and 15° in the XRD figure? The mineral components other than Heulandite need to be explained.

Added additional minerals to XRD figure

6) In the conclusion part, the optimum dosage of natural zeolite and sodium-calcium glass powder mixture should be put forward considering the strength, workability and hydration reaction of cement mortar. 

The conclusions were reviewed, corrected and supplemented with the results obtained.

7) It should mention the scope for further research as well as the implications of the study. I recommend including the limitations regarding the consideration of damage indicated in this review in the limitations assessment. This part of the document can be improved and completed with more rigour. The advantages of the use of this mixture to the performance of cement mortar should be mentioned briefly.

This study investigates the complex effect of zeolite and glass powder mixture on its porosity parameters. The data presented in Figure 7 show that using these materials separately and replacing cement with up to 20% zeolite or glass increases the open porosity, while replacing 25% cement with a mixture of these minerals has achieved a slight but positive improvement in porosity parameters.

Thank you for your time and valuable comments.

Reviewer 5 Report

General comment:

This paper investigated the effect of the mixture consisting of natural zeolite (clinoptilo- lite) and soda lime glass powder on the hydration, mechanical properties and open porosity of cement concrete. The effect of mineral materials on the technological, physical-mechanical proper- ties and open porosity of the mortar was investigated by increasing the content of natural zeolite or glass powder added to the mix by weight of cement in increments of 5 %. The microstructure of hardened cement paste modified with the aforementioned mineral additions was analyzed by means of mechanical, microcalorimetry, X-ray diffraction tests and thermogravimetric analysis. The optimal formulation of the cement paste, and particle size distribution were deter- mined to achieve lower open porosity irrespective of the cement dilution effect and the increase of W/C ratio up to 0.67. The content of the article is comprehensive, but some details are not well done. There are some comments below that authors may consider revising their paper.

Technique comments:

(1)Page 3, Line 124 and line 130,Table 1 has the same name as Table 2.

(2)Page 8, Line 270-271,Figure 3 and Figure 4 have no legend description of blue lines. What does the blue line mean?

(3)Page 10, Line 332-333, the authors stated that the results of these tests are presented in Figure 7. The results of compressive strength after 28 days given in Fig. 7. According to the content of the article, it should be Figure 6.

(4)There is one paper that the authors may consider for added in the introduction section (Assessment of Automatic Induction Self-Healing Treatment Applied to Steel Deck Asphalt Pavement. Automation in Construction. https://doi.org/10.1016/j.autcon.2021.104011).

(5)Page 11, Line 348-350, Figure 7(b) lacks legend description.

(6)Page 12, Line 372-373, Figure 9 should be behind the picture.

(7)Page 14, Line 410-411, the authors demonstrated that minor endothermic effects at approx. 190 º C and 230 C temperature marked by 410 dotted lines in Fig. 11 should be also noted. However, where is Figure 11?

Author Response

Dear Reviewer,

First of all, I would like to apologize for the inconvenience of reading our article, as it was converted to an older version of Word at the last minute and the format has changed as a result.

According your comments and suggestions:

(1)Page 3, Line 124 and line 130,Table 1 has the same name as Table 2.

Changed

(2)Page 8, Line 270-271,Figure 3 and Figure 4 have no legend description of blue lines. What does the blue line mean?

It means CEM+GP10, changed

(3)Page 10, Line 332-333, the authors stated that the results of these tests are presented in Figure 7. The results of compressive strength after 28 days given in Fig. 7. According to the content of the article, it should be Figure 6.

In Figure 7 presented results of mortars No 2, in Figure 6 results of mortars No 1, left as is

(4)There is one paper that the authors may consider for added in the introduction section (Assessment of Automatic Induction Self-Healing Treatment Applied to Steel Deck Asphalt Pavement. Automation in Construction. https://doi.org/10.1016/j.autcon.2021.104011).

Maybe if the article has things to do with this study

(5)Page 11, Line 348-350, Figure 7(b) lacks legend description.

Corrected

(6)Page 12, Line 372-373, Figure 9 should be behind the picture.

Agree, corrected

(7)Page 14, Line 410-411, the authors demonstrated that minor endothermic effects at approx. 190 º C and 230 C temperature marked by 410 dotted lines in Fig. 11 should be also noted. However, where is Figure 11?

Changed to Fig. 10

Thank you for your time and valuable comments.

Round 2

Reviewer 1 Report

Personally, I am not really convinced by most of the responses of the authors to my questions. Sometimes even the authors are repeating the same answer (copy-paste) to more than one question. 

The second major issue; the authors submitted a revised version of the manuscript in plain text! How I can detect and follow up with the changes embedded in the text.

The authors, as we all do in such a case, have to submit a highlighted text where the modifications and changes should be highlighted with a different color. 

This is the only way I can make sure that what I requested as changes are implemented. 

Author Response

Dear reviewer,

We regret that our answers seems unconvincing or superficial for you. We have tried to respond briefly to your comments, and now of the changes based on your comments have been made in the attached file.

The main changes:

  • adjusted abstract;
  • the introduction was supplemented by data from articles on the effects of zeolite or ground glass on the properties of cement based composites, thus adding data from studies in which part of the cement is replaced by a mixture of ground glass and rice husks ashes;
  • the methodology was supplemented with detailed methods for testing the mechanical strength of mortars, and the algorithm for calculating the porosity of a fresh mixture was explained.
  • part of the results and discussions section were supplemented or adjusted by the authors' comments on the properties of the fresh mortar; the results of the mechanical strength test; XRD test results; TGA test results;
  • reformulated research findings and conclusions;
  • corrected, supplemented or added pictures;
  • minor adjustments were made based on the reviewers' comments.

Details of the changes can be viewed in the attached file with the highlighted changes.

Respectfully,

Dalius Kriptavicius

Reviewer 2 Report

Dear Authors

Please highlight changes on the manuscript. 

Author Response

Dear reviewer,

The main changes:

  • adjusted abstract;
  • the introduction was supplemented by data from articles on the effects of zeolite or ground glass on the properties of cement based composites, thus adding data from studies in which part of the cement is replaced by a mixture of ground glass and rice husks ashes;
  • the methodology was supplemented with detailed methods for testing the mechanical strength of mortars, and the algorithm for calculating the porosity of a fresh mixture was explained.
  • part of the results and discussions section were supplemented or adjusted by the authors' comments on the properties of the fresh mortar; the results of the mechanical strength test; XRD test results; TGA test results;
  • reformulated research findings and conclusions;
  • corrected, supplemented or added pictures;
  • minor adjustments were made based on the reviewers' comments.

Details of the changes can be viewed in the attached file with the highlighted changes.

Respectfully,

Dalius Kriptavicius

Reviewer 3 Report

Thanks to the authors' hard work. Even though not all my comments are addressed, I believe that is due to a lack of data. Hope the authors could solve these issues in the following studies. No further review process is needed.

Author Response

Dear reviewer,

Thank you for your time, attention, and comments that have allowed us to improve our article.

Respectfully,

Dalius Kriptavicius

Reviewer 5 Report

The comments have been addressed, the introduction could be improved with some new publications related to sustainable materials.  (eg. Investigation of the mechanical and shrinkage properties of plastic-rubber compound modified cement mortar with recycled tire steel fiber; The Performance Evaluation of Asphalt Mortar and Asphalt Mixture Containing Municipal Solid Waste Incineration Fly Ash; The performance of micropore-foamed geopolymers produced from industrial wastes)

Author Response

Dear reviewer,

In attached file you may see highlighted part of introduction which was changed according you suggestions.

Respectfully,

Dalius Kriptavicius

Round 3

Reviewer 1 Report

The issue of novelty and contribution of this paper compared to existing studies in the field was not addressed

There is a great issue with Table 2! Please address it

Lines 316-318: According to this statement, the fresh porosity was theoretically calculated not experimentally tested!!

Author Response

Dear reviewer,

The scientific novelty of the article consists of the following parts:

  1. The addition of natural zeolite increases the water demand of its absorbent properties, so sodium lime glass was added and this study examines the complex effect of these additions on the cement hydration, technological and mechanical properties of cement based composites.
  2. Was investigated the influence of the fineness of natural zeolite on hydration reactions was studied by mixing zeolite ground to a very small particle size in the form of an aqueous suspension.
  3. The effect of complex additive on hydration products was investigated and it was found that the mineral composition of modified cementitious stone is different in comparison to control samples; ettringite is not determined in the modified mixtures and calcium aluminosilicate hydrates such as katoite and kamaishilite are found.

Table 2 is supplemented with additional physical parameters.

Porosity of fresh mortars was calculated theoretically because there are no testing methods how to measure entrapped or entrained air content. We measured density of fresh mortar then calculated what would be the density if there were no voids in the mixture and then calculated the percentage difference between fact and maximum possible.

Respectfully,

Dalius Kriptavicius

Reviewer 2 Report

This manuscript can be accepted in current form.

Author Response

(The authors gave the same response as above.)
